# Enhancing a Natural Killer: Modification of NK Cells for Cancer Immunotherapy

**DOI:** 10.3390/cells10051058

**Published:** 2021-04-29

**Authors:** Rasa Islam, Aleta Pupovac, Vera Evtimov, Nicholas Boyd, Runzhe Shu, Richard Boyd, Alan Trounson

**Affiliations:** 1Cartherics Pty Ltd., Clayton 3168, Australia; rasa.islam@hudson.org.au (R.I.); aleta.pupovac@hudson.org.au (A.P.); vera.evtimov@hudson.org.au (V.E.); nicholas.boyd@hudson.org.au (N.B.); runzhe.shu@hudson.org.au (R.S.); richard.boyd@hudson.org.au (R.B.); 2Department of Obstetrics and Gynaecology, Monash University, Clayton 3168, Australia

**Keywords:** natural killer (NK) cells, pluripotent stem cells, allogeneic immunotherapy, cancer

## Abstract

Natural killer (NK) cells are potent innate immune system effector lymphocytes armed with multiple mechanisms for killing cancer cells. Given the dynamic roles of NK cells in tumor surveillance, they are fast becoming a next-generation tool for adoptive immunotherapy. Many strategies are being employed to increase their number and improve their ability to overcome cancer resistance and the immunosuppressive tumor microenvironment. These include the use of cytokines and synthetic compounds to bolster propagation and killing capacity, targeting immune-function checkpoints, addition of chimeric antigen receptors (CARs) to provide cancer specificity and genetic ablation of inhibitory molecules. The next generation of NK cell products will ideally be readily available as an “off-the-shelf” product and stem cell derived to enable potentially unlimited supply. However, several considerations regarding NK cell source, genetic modification and scale up first need addressing. Understanding NK cell biology and interaction within specific tumor contexts will help identify necessary NK cell modifications and relevant choice of NK cell source. Further enhancement of manufacturing processes will allow for off-the-shelf NK cell immunotherapies to become key components of multifaceted therapeutic strategies for cancer.

## 1. Introduction

Immunotherapies utilizing T cells expressing chimeric antigen receptors (CARs) have shown encouraging clinical success in blood cancers. However, the time and difficulty of retrieving sufficient numbers of patient T cells, the expense of manufacture, the limited success with solid tumors and patient side effects remain significant roadblocks [1]. Recently, allogeneic natural killer (NK) cells have emerged as alternative immunotherapeutic agents offering unique advantages compared to T cells. NK cells naturally express multiple cancer recognition receptors, do not induce graft-versus-host disease (GvHD) and have a reduced potential for inducing adverse life-threatening events such as a cytokine storm in patients [1]. However, their clinical translation poses its own challenges including logistical difficulty in obtaining sufficient numbers for treatment, difficulties in genetic engineering to enhance function and their limited persistence in vivo [1,2]. To overcome these issues, NK cells are being derived from stem cell sources such as induced pluripotent stem cells (iPSCs) which possess limitless self-renewal potential and can be genetically engineered to increase NK cell killing specificity, potency and efficacy in vivo and subsequently in the clinic [3]. Clustered regularly interspaced short palindromic repeats (CRISPR)/Cas9 technology has been invaluable for the genetic engineering of iPSCs, and has the potential for successful genetic editing of NK cell receptors themselves. Accordingly, this technology is evolving to support the creation of large numbers of gene-edited, functionally “supercharged” NK cells [3,4]. Other ways to increase the effect of NK-based therapy include adjuvant inhibitory checkpoint therapy or application of synthetic compounds to increase the yield and potency of these effector cells [5].

This review aims to summarize key biological characteristics of NK cells, the current strategies and hurdles in modification of NK cells, as well as the current and future clinical utilization of modified or genetically edited NK cells in addressing these issues.

## 2. NK Cell Biology

### 2.1. NK Cells in Innate and Adaptive Immunity

NK cells are classically considered to be innate immune effector lymphocytes due to their lack of antigen-specific receptors. However, more recently, they are known to contribute to both arms of the immune system through their regulatory functions exerted by cytotoxicity and cytokine production [6]. These cells, accounting for approximately 10% of human peripheral blood (PB) lymphocytes, primarily develop in the bone marrow and other secondary lymphoid tissues including the tonsils and spleen. NK cells predominantly circulate in PB with some tissue-specific subsets residing, at least temporarily, within the liver, intestine, spleen and bone marrow. They can also be found in the uterus during pregnancy, in the lungs and the skin [6,7]. Generally, circulating NK cells can be subdivided according to the surface expression of CD56 and CD16 [8]. These subsets exhibit major functional differences in their cytotoxicity, cytokine production, and homing capabilities and their functions are governed by activating, co-stimulatory and inhibitory receptors. Generally, CD56^bright^ NK cells are predominantly found in tissues and have poor cytolytic activity, while CD56^dim^ NK cells are found in PB, have stronger cytolytic activity and express CD16, the molecule responsible for initiating antibody dependent cell-mediated cytotoxicity (ADCC) (Figure 1) [8]. Further mechanistic understanding of major PB NK subsets is key to unravelling the complex immunosurveillance they orchestrate to transition from exerting innate to adaptive immune responses.

### 2.2. NK Cell Receptors and Their Role in Tumor Surveillance

NK cell regulation is a dynamic integration of simultaneously transduced signals from inhibitory, activating, cytokine and adhesion receptors in response to “altered self” cells such as tumor cells or stressed cells. Normally, self-peptides presented within major histocompatibility complex (MHC) molecules, specifically human leukocyte antigens (HLA) in humans are recognized by the immune system as “safe” and are thus ignored. This is termed self-tolerance. In the “missing self” model, NK cells are activated when their inhibitory receptors fail to recognize target cells due to an incomplete or incompatible set of “self-identifier” (HLA) molecules [9]. NK inhibitory receptors fall into two broad subcategories—the monomeric type I glycoproteins of the immunoglobulin (Ig) superfamily known as killer cell Ig-like receptors (KIRs) and the type II glycoprotein with a C-lectin scaffold, known as NKG2A (Figure 1) [10]. KIRs ligate with HLA types A, B or C, but mainly with HLA-C. These are highly polymorphic molecules resulting in a variegated KIR repertoire. In contrast, NKG2A engages with the highly conserved HLA-E that has limited polymorphism. NKG2A expression also precedes KIRs in NK cell ontogeny, suggesting that KIR expression marks mature NK cells [11].

Infected, foreign or cancer cells exchange normal self-peptides for “non-self” peptides, enabling targeting by the immune system, with malignant transformation commonly enabling CD8^+^ cytotoxic T cell-mediated lysis [9]. However, tumor cells often escape this immunosurveillance, by downregulating the surface expression of MHC-I molecules which by-passes CD8^+^ T cell killing. To counter this, the lower MHC expression disengages host NK cell inhibitory receptors and activates them, enabling cancer killing [8,10]. This underpins the importance of NK cells in cancer immunotherapy. The “induced self” model postulates that transformed cells (abnormal cells such as tumors) express a variety of stress-related ligands such as Fas, death receptor 5, MHC I chain-related proteins MICA/MICB, heat shock proteins and UL16 binding proteins, which are able to activate NK cells via their activating receptors [8,12]. NK-activating receptors can be categorized into type II C-type lectin-like molecule termed NKG2D and type I transmembrane proteins belonging to the Ig superfamily natural cytotoxicity receptors (NCRs) that include NKp46, NKp30 and NKp44 receptors (Figure 1) [12]. Each type of receptor may have a distinct role in NK cell development, with discrete expression of NKG2D proteins observed prior to NCR expression in the development pathway [11].

Given the dynamic roles of NK cells in tumor surveillance, the use of NK cells as a next-generation tool for adoptive immunotherapy is gaining significant traction. It is well established that NK cells demonstrate strong in vivo and in vitro tumoricidal effects [10] and NK cells in preliminary clinical trials have demonstrated better safety profiles than edited T cells such as CAR-T cells [13]. Unlike T cells, NK cells do not react against HLA-mismatch, negating the likelihood of GvHD. The lack of clonal expansion and secretion of interleukin (IL)-3 and granulocyte-macrophage colony stimulating factor (GM-CSF) on NK cell activation makes cytokine release syndrome (CRS) unlikely, since the condition is mostly attributed to interferon (IFN)-γ, tumor necrosis factor (TNF)-α, and IL-6 secreted by activated T cells and IL-1 from macrophages [14]. Such advantages have led to increased utilization of NK cells in clinical studies. Twenty-six clinical trials were performed in the last 10 years involving NK cells, with an increasing number of current trials assessing engineered NK cell products [15].

## 3. Modifying NK Cells to Be Better Effector Cells

A number of strategies can be employed to increase the therapeutic efficacy of NK cells, including the use of cytokines (Section 3.1), targeting inhibitory checkpoints (Section 3.2), the use of synthetic compounds and recombinant proteins (Section 3.3), addition and manipulation of CARs (Section 3.4) and the genetic ablation of inhibitory molecules (Section 3.5).

### 3.1. Cytokine-Based Cell Expansion, Propagation and Therapies

The cytokines IL-2, IL-15, IL-12, IL-18, and IL-21 all regulate NK cell activation, maturation and survival to differing degrees. The administration of cytokines in vivo or the pre-treatment of NK cells before adoptive transfer have been used to improve NK cell performance [16]. Largely, IL-2 was most commonly used and was approved first for clinical use. However, the clinical response of IL-2 against solid tumors in particular, has been ineffective. In clinical trials using ex vivo IL-2 pre-treated autologous NK cells, as well as the simultaneous administration of IL-12 and autologous NK cells did not induce clinical improvement in patients with metastatic digestive tumors or breast cancer [17,18]. This may be attributed to impaired functional responses of NK cells taken from the cancer patients. Allogeneic NK cells have provided better responses particularly in hematological cancer. Furthermore, ex vivo IL-2-expanded allogeneic NK-92 cell treatment resulted in better clinical response in most patients with advanced lung cancer [19]. However, a major setback to the use of IL-2 with NK cells is the stimulation of T regulatory cells which compete for IL-2 [20,21]. When compared to IL-2, IL-15 is thought to have a more potent effect on NK cell expansion [22] and enhances NK cell cytotoxicity as shown in several preclinical studies [23,24]. Some of the antitumor effects have been in part attributed to NKG2D activity [25]. While it has been suggested that soluble IL-15 does not appear to expand T regulatory cells [16], IL-15 increases expression of CD25 and FOXP3 in peripheral CD4^+^CD25^-^ T cells in the absence of antigen stimulation, cells which are similar to conventional T regulatory cells [26]. Several clinical trials have shown good outcome and feasibility for the use of IL-15 with NK cells for several cancer types including acute myeloid leukemia, advanced non-small-cell lung cancer and pediatric refractory solid tumor [27,28,29,30]. However, there is evidence to suggest that human NK cells which are continuously treated with IL-15 undergo a process consistent with exhaustion [31].

The IL-15 cytokine analogs N-803 (also known as ALT-803) [32], P22339 [33] and NKTR-255 [34] have better biological activity including potency and half-life than their IL-15 counterpart and promote NK cell function and antitumor activity. N-803 inhibits complement activation and increases the half-life and stability of NK cells through an Fc domain which mediates more ideal pharmacokinetics with prolonged cytokine function [35]. This super agonist potently enhances the killing capacity of NK cells against ovarian cancer cell lines (OVCAR-3, MA-148, OVCAR-5, SKOV3, and A1847) with significant increases observed in CD107a, IFN-γ and TNF-α expression depending on the cell line targeted [36]. N-803 treatment of NK cells from patient ascites significantly increased degranulation and IFN-γ production against K562 targets. Additionally, in an in vivo model, only animals treated with intraperitoneal N-803 displayed an NK-dependent significant decrease in tumor size [36]. Therefore, N-803 could be an ancillary pre-treatment to increase the cytotoxicity of effector NK cells. N-803 was found to be safe in clinical trials for advanced solid tumors and several trials using this analog are still ongoing [37,38]. NKTR-255 is also in clinical trials currently recruiting for patients with relapsed/refractory multiple myeloma (MM) and non-Hodgkin lymphoma (NHL) (NCT04136756).

Membrane IL-21 alone [39] and IL-21 in combination with IL-15 [40,41,42] promotes the expansion of NK cells ex vivo with preclinical studies establishing efficacy against solid tumors. Several clinical trials also show the value of using IL-21 alone against metastatic melanoma, but drug combinations have yielded mixed outcomes [43,44]. A phase I clinical trial which carried out multiple infusions of membrane-bound (mb)IL-21 ex vivo-expanded donor-derived NK cells on patients with myeloid malignancies demonstrated safety and retained low relapse rates [45]. Other clinical studies using IL-21-expanded NK cells are currently still ongoing (NCT02809092).

In studies using murine models, the antitumor efficacy of IL-12 has been substantial. However, clinical trial outcomes show mixed responses and clinical benefits have been modest. Early phase clinical trials have shown safe administration with antitumor monoclonal antibodies (mAbs). Immunocytokines are tumor-specific antibodies fused to a cytokine which facilitate delivery of the cytokine to the tumor microenvironment (TME). Thus, fusion of IL-12 or other cytokines to an antitumor mAb may be beneficial for use in combination therapies targeting resistant tumors, solid tumors or for linking to bi-specific or tri-specific killer cell engagers (BiKEs or TriKEs) [46]. In this regard, since systemic cytokine administration may cause severe toxicities, limiting its use as an immunotherapeutic agent, immunocytokines may overcome some of these limitations. Studies have shown that treatment with immunocytokines leads to the targeted increase in the density of NK cells and lymphocytes in the tumor extracellular matrix [16,46]. For example, NK cells expressing mbIL-15 show superior proliferation and survival and have higher cytotoxicity against tumor cells in vitro and in vivo, compared to NK cells without mbIL-15 [47,48,49]. Thus, genetic engineering of NK cells, and in particular CAR-NK cells to express IL-15 can enhance the persistence and cytotoxicity against tumor cells. Additionally, CAR-NK cells co-expressing mbIL-15 have higher expansion (more than 4-fold) without feeder cells than CAR-NK cells without mbIL-15 expression [50]. Engineering of CAR-NK cells with on-board cytokines such as IL-15, is laying foundations for new therapeutic options to improve clinical efficacy. This is exemplified by several products incorporating IL-15 (Table 1) and clinical trials using CAR-NK cells with on board IL-15 targeting B cell tumors (NCT03056339 and NCT04245722).

Like IL-12, the in vivo antitumor efficacy of IL-18 is well established preclinically, but very few clinical studies are evaluating its safety and efficacy in patients [16]. Thus, IL-18 may be useful in future combination therapies. In this regard, stimulation of NK cells with a combination of IL-12, IL-15, and IL-18 bestowed memory like effector responses after re-stimulation of NK cells. These are known as cytokine-induced memory-like NK cells, which may be a promising tool not only for cancer immunotherapy [51,52] but for other diseases too. A recent study has used this approach to generate memory like NK cells armed with an anti CD19 CAR [53]. These cells exhibited enhanced responses against primary lymphoma cells and their autologous lymphoma in vitro. Furthermore, they controlled lymphoma burden and improved survival in human xenograft models [53].

**Table 1 cells-10-01058-t001:** Products in development using CAR-NK cells with onboard IL-15 [54].

Product	Company	Description
GoCAR™	Bellicum Pharmaceuticals	CD123 or HER2 CAR-NK cells with rimiducid-inducible iMC and autocrine IL-15 exhibit enhanced persistence and antitumor activity in CD123^+^ AML and HER2^+^ solid tumor models.
FT596	Fate Therapeutics	CD19 CAR with CD16 Fc receptor and IL-15 receptor fusion protein with a flexible IL-15 increases NK cell activation and targets both CD19 and CD20 in B cell malignancy.
NKX-101	Nkarta	NKG2D CAR-NK cells and mbIL-15 for further activation as well. NKG2D ligands are expressed in many tumor cells.
TAK-007	Takeda	CD19 CAR-NK cells with a CD28 costimulatory domain, IL-15 and an inducible caspase 9 suicide gene.

Abbreviations: AML, acute myeloid leukemia; CAR, chimeric antigen receptor; CD, cluster of differentiation; HER2, human epidermal growth factor receptor 2; IL-15, interleukin-15; iMC, MyD88/CD40; mbIL-15, membrane-bound interleukin-15; NK, natural killer; NKG2D, transmembrane protein belonging to the NKG2 family of C-type lectin-like receptors.

### 3.2. Therapeutic Approaches to Enhancement of NK Cell Function

The success of inhibiting the immune checkpoints cytotoxic T-lymphocyte-associated antigen 4 (CTLA-4) and programmed death-1 (PD-1) in T cell-based immunotherapy has directed focus on immune checkpoint targeting in NK cells. Besides innate NK cell receptors such as KIRs and NKG2A, CTLA-4 and PD-1 are also being targeted to improve NK cell function (Figure 2). Significantly, many current therapies are targeting T cell immunoreceptors with Ig and ITIM domains (TIGIT) and CD96, exemplifying the importance of these molecules in NK cell function (Figure 2). Other molecules targeted include T cell Ig and mucin-containing domain-3 (TIM-3), CD200/CD200 receptor (CD200R), lymphocyte activation gene-3 (LAG-3), OX40/OX40L and IL-1 receptor 8 (IL-1R8). Therapies targeting these checkpoints molecules may reverse dysfunctional NK cells in the tumor setting, as well as assist in complementing T cell immunotherapies [55].

#### 3.2.1. KIR

Anti-KIR antibodies disrupt HLA-KIR matching, thereby limiting self-recognition of the HLA by NK cells and allowing for the maximum tumoricidal effect [56]. Lirilumab (IPH2102) is a human anti-KIR mAb targeting the KIR2DL1/2/3 NK inhibitory receptors (Figure 2). It is in clinical evaluation and development as a stand-alone or combinatorial therapy with other anticancer immunomodulatory drugs such as lenalidomide, mAbs such as elotuzumab and rituximab, or the immune checkpoint inhibitor mAbs, ipilimumab and nivolumab. Phase I trials have indicated that lirilumab, in combination with lenalidomide, may be a more promising therapy for MM patients than lirilumab alone [57]. Similarly, preclinical studies indicating functional augmentation of lirilumab by elotuzumab for MM have paved the way for an ongoing phase I clinical trial (NCT02252263). Lirilumab used in combination with rituximab is also undergoing clinical trials in patients with relapsed, refractory or high risk chronic lymphocytic leukemia (CLL) (NCT02481297). This combination was shown to increase NK cytotoxicity in a human cell line model of lymphoma, providing an approach to potentiate therapeutic benefit of antitumor antibodies that mediate ADCC [58]. A recent anti-KIR3DL2 mAb, lacutamab or IPH4102, is undergoing assessment in a phase II clinical trial (NCT03902184) as a single agent or in combination with chemotherapy for the treatment of T cell lymphomas (Figure 2). KIR blockade displayed limited side effects in these clinical studies [59].

Neither anti-KIR single blockade (NCT01687387), nor combined with anti-CTLA-4 (NCT01750580), displayed better efficacy than anti-CTLA-4 alone. However, combination blockade of KIR and PD-1 showed increased objective response rate for advanced head and neck tumor patients previously treated with chemotherapy (NCT01714739) [60]. Additionally, preliminary data from KIR blockade in combination with 5-azacytidine, a DNA methyltransferase inhibitor, as a therapy for refractory or relapsed acute myeloid leukemia (AML) showed that 20 percent of the first 25 patients responded to therapy, with two patients achieving complete remission (NCT02399917). Therefore, these preliminary data have encouraged more ongoing clinical trials utilizing KIR blockade alone or in combined therapies (e.g., combined with anti-CD20 in NCT02481297, with anti-PD-1, and 5-azacytidine in NCT02599649, or with anti-SLAMF7 in NCT02252263) [60].

#### 3.2.2. NKG2A

Non-classical MHC-I, HLA-E, is the ligand of NKG2A in human NK cells [61]. HLA-E is widely expressed by a variety of tumors (e.g., lung, pancreas, stomach, colon, head and neck, and liver) and binding of NKG2A to its ligand suppresses the effector functions of T and NK cells [62,63]. Monalizumab (IPH2201), a humanized mAb against NKG2A has shown improvement of NK cell function in vitro and is well tolerated in patients with gynecological malignancies (Figure 2) [63]. This antibody is also being evaluated in combination with the anti PD-1 mAb durvalumab, the epidermal growth factor receptor (EGFR) inhibitor cetuximab and the Bruton tyrosine kinase inhibitor, ibrutinib [62]. The monalizumab-durvalumab combination has significant antitumor efficacy in vitro, along with complete response rate and a manageable toxicity profile in a clinical setting (NCT02671435) [64]. The monalizumab-cetuximab combination administered to patients with head and neck squamous cell carcinoma also showed encouraging objective response rate, progression-free survival and overall survival [65]. mAbs that block such ligation could therefore potentiate NK antitumor activity by preventing NK cell exhaustion. Along with the MHC-I inhibitory NK cell receptors, numerous other immune checkpoints are involved in NK cell impairment in cancer and other diseases as shown by the aforementioned combination blocking success. These are outlined below.

#### 3.2.3. CTLA-4

CTLA-4 is expressed on T cells and competes with CD28, a costimulatory receptor, for ligands CD80 and CD86 on cancer cells [66]. Ipilimumab blockade of CTLA-4 has been shown to increase CD4^+^ T cells, IL-2 production and cause FcR-dependent depletion of regulatory T cells: all of which indirectly augment NK cell activation and degranulation (Figure 2) [67]. Blockade with this antibody in a tumor context has also induced ADCC and TNF-α release by NK cells [68]. Different subsets of NK cells have also been associated with either positive or negative clinical outcomes to CTLA-4 blockade. The CD56^bright^/CD16^dim^ NK cell subset in particular was associated with the regression of pancreatic cancer after treatment with ipilimumab [69], indicating the importance of functional characterization of different NK cell subsets in different cancers. However, CTLA-4 is only expressed at very low levels in healthy NK cell cytoplasm, and therefore might not be an adequate target for direct checkpoint inhibition for NK cells as it is for T cells [70,71].

#### 3.2.4. PD-1/PD-L1

PD-1 is upregulated on NK cells in several cancers including ovarian, Kaposi sarcoma, renal cell carcinoma, digestive cancers and MM, with the degree of expression varying from cancer to cancer. This upregulation suggests a functional dysregulation in NK cells, exemplified by functionally exhausted PD-1^+^ NK cells [72]. Several murine models have also shown that ligation of PD-1 on NK cells with its ligand programmed death-ligand 1 (PD-L1) on tumor cells can exhaust NK cells and strongly suppress their antitumor effect [73]. Combination therapy with the anti-PD-1 mAb, nivolumab has been used to reconstitute NK cell function in patients with solid tumors (Figure 2) [74] and the anti PD-L1 mAb, avelumab significantly enhances NK cell-mediated cytotoxicity against triple-negative breast cancer cells (TNBC) [73]. Tumor cells expressing higher levels of PD-L1 were also found to be more sensitive to avelumab-mediated ADCC. Combined therapeutic strategies using histone deacetylase inhibitors results in enhanced NK cell tumor cell lysis and avelumab-mediated ADCC in multiple carcinoma cell types [75]. Therefore, targeting the PD-1/PD-L1 pathway may improve NK-mediated tumor cell elimination in TNBC [73]. However, the role of the PD-1/PD-L1 axis in NK cell antitumor function seems to be TME dependent and complex. Immune evasion via this pathway on NK cells is more prominent in Hodgkin’s lymphoma compared to diffuse large B cell lymphoma due to the skewing toward an exhausted PD-1-enriched NK cell phenotype as well as the indirect suppression of NK cells via PD-L1/PD-L2-expressing tumor-associated macrophages [76].

#### 3.2.5. TIGIT and CD96

TIGIT, which is highly expressed on T cells and NK cells, exerts inhibitory function through its intracellular ITIM and immunoreceptor tyrosine tail (ITT)-like domain [70]. Along with its co-inhibitory receptor CD96 (TACTILE), TIGIT can bind to ligand CD155, also known as the poliovirus receptor (PVR), which is upregulated in many cancers. The ligation of TIGIT to CD155 results in suppressed cytolytic activity and IFN-γ production of NK cells in vitro, possibly via the NF-κB pathway [77,78]. Further, NK cells from TIGIT overexpressing transgenic mice generate less IFN-γ after TIGIT/PVR ligation whereas those from TIGIT KO mice produce more exemplifying this suppression in vivo. TIGIT expression, but not PD-1 or CTLA-4, was found to be upregulated on exhausted NK cells and was associated with tumor progression in severe combined immunodeficient mice [79]. Blockade of TIGIT in these mice prevented exhaustion and promoted not only NK cell-dependent tumor immunity, but also tumor-specific T cell function in an NK cell-dependent manner, resulting in prolonged mouse survival [79].

In vivo data have also demonstrated that CD96 is an important checkpoint for NK cell effector functions. CD96^−/−^ mice are highly susceptible to induced innate inflammation, but display more resistance to chemically induced fibrosarcoma or experimental metastases [80]. Enhanced IFN-γ production by NK cells in CD96^−/−^ mice has been observed, whereas binding of CD96 to its ligand reduced IFN-γ production [80]. Interestingly, there was no difference observed between CD96^−/−^ and wild-type NK cells in their cytotoxicity in vitro, indicating a regulatory effect of CD96 on NK cytokine production rather than other direct modes of cytotoxicity.

Dysregulation of the TIGIT/CD96/CD226/CD155 axis and NK cell function has been observed in certain cancers. In hepatocellular carcinoma patients, the percentage and intensity of CD96 on NK cells, as well as the numbers of CD96^+^ NK cells, were higher in tumor-infiltrating NK cells compared with NK cells from peri-tumoral tissues [81]. It was observed that while TIGIT expression was similar between pancreatic cancer patients to that of healthy controls, CD155 was upregulated in pancreatic cancer tissue. The lower expression of CD226 and CD96 on NK cells may be contributing to tumor escape [82]. Functionally exhausted CD96^+^ NK cells were significantly increased in the tumor tissues of patients with worse prognosis of hepatocellular carcinoma. These cells had impaired IFN-γ and TNF-α production and high gene expression of inhibitory cytokines IL-10 and transforming growth factor-beta 1 (TGF-β1). Blocking the CD96-CD155 axis or TGF-β1 restores NK cell function by reversing NK cell exhaustion, suggesting a possible therapeutic role of CD96 in fighting liver cancer [81]. Co-blockade of CD96 and PD-1 potently inhibits lung tumor metastases and increases local NK cell IFN-γ production and infiltration [83]. Furthermore, when combined with CD96 blockade, anti-CTLA-4, anti-PD-1, or chemotherapy were also more effective.

Four mAbs against TIGIT are currently under investigation for immunotherapeutic use (Figure 2 and Table 2). Tiragolumab (also known as MTIG7192A and RG6058) is a fully human anti-TIGIT antibody developed by Genentech/Roche that is being analyzed for use in immunotherapy in preclinical studies in combination with antibodies to PD-1. Bristol-Myers Squibb also initiated a phase I/II study with an anti-TIGIT antibody (BMS-986207) as monotherapy or in combination with nivolumab in advanced solid tumors (NCT02913313) (Figure 2 and Table 2). Etigilimab (OMP-313M32), developed by OncoMed Pharmaceuticals is under study in a phase I clinical trial (NCT031119428) in solid tumors as a monotherapy or in combination with anti-PD-1. AB154, developed by Arcus Bioscience is in a phase I clinical trial (NCT03628677) that is evaluating the safety, pharmacokinetics, pharmacodynamics and preliminary efficacy in advanced solid tumors as monotherapy or combined with AB122 (anti-PD-1) (Figure 2 and Table 2) [84]. In terms of clinical efficacy, it is crucial to note that TIGIT is upregulated in NK cells, which might indicate that unlike other checkpoint molecules such as PD-1, CTLA-4 or LAG-3, which are more restricted to expression on tumor-infiltrating T cells, TIGIT is a checkpoint receptor more specific to NK cells.

#### 3.2.6. TIM-3

Among resting lymphocyte populations, NK cells have the highest percentage expression of TIM-3 [85]. Galectin-9 is a ligand for TIM-3 and is upregulated in a variety of cancers [86]. TIM-3 has been described as both an activation marker and activation limiter of NK cells, as galectin-9 engagement can have contrasting effects [86]. For example, it has been shown that TIM-3 is upregulated on human NK cells after activation and promotes IFN-γ production in response to galectin-9 [85]. However, it was also demonstrated that when TIM-3 was crosslinked on NK cells with antibodies or encountered target cells that expressed galectin-9, it suppressed NK cell-mediated cytotoxicity [86]. The TIM-3-galectin-9 pathway acts to suppress anticancer immune surveillance, contributing in particular to the immune escape of AML cells [87]. TIM-3 is therefore an emerging target for AML. It may be effective to target TIM-3 in patients displaying resistance to PD-1 blockade, particularly in cases where TIM-3 expression is increased and thought to drive suppression [88]. A phase I clinical trial testing combinational treatment with PD-1 blockade and a TIM-3-blocking therapeutic antibody, TSR-022 (developed by Tesaro Inc), is currently being administered in patients with advanced tumors (Figure 2) [70]. TSR-022, is under investigation in three clinical trials for advanced cancer alone or in combination with anti-PD-1 (NCT02817633, NCT03307785), and in primary liver cancer (NCT03680508) in combination with anti-PD-1. A Novartis anti-TIM-3 antibody, MBG453 is also being evaluated as monotherapy or in combination with anti-PD-1 in patients with advanced malignancies (NCT02608268) and patients with AML or high risk myelodysplastic syndromes (NCT03066648) (Figure 2) [84]. However, the possibility remains that only galectin-9-positive tumors may be more susceptible to NK cell recognition and targeting via interaction with TIM-3, which may be a potential limitation of targeting this molecule [85].

#### 3.2.7. CD200R

Upregulation of the immunosuppressive cell surface glycoprotein, CD200, is a common feature of AML and is associated with poor patient outcomes. CD200^hi^ patients display a 50% reduction in the frequency of activated NK cells (CD56^dim^CD16^+^) and a significant reduction in activating receptor surface expression and IFN-γ levels compared to that of CD200^lo^ patients [89]. As such, CD200 has a direct and significant suppressive influence on NK cell activity in AML patients and may contribute to the increased relapse rate in CD200^+^ patients, and therefore the CD200R is an attractive target to block, particularly for tumors with high CD200 expression [89]. However, blocking CD200R on NK cells may be difficult as it is also expressed on CD3^+^ T cells [90], and reproducibly at low levels on CD3^−^CD56^+^ NK cells [91].

Other novel immune checkpoints are also being investigated in NK cells, including LAG-3 which binds MHC-II and is expressed on ~10% of NK cells. However, since NK cells do not generally interact with MHC-II, alternative ligands have been proposed for LAG-3 such as liver sinusoidal endothelial cell lectin expressed on melanoma cells, and galectin-3, expressed on stromal cells commonly found in the TME [92]. However, how relevant they are for NK cells remains to be investigated and the significance of LAG-3 expression in NK cells remains largely unclear [70]. Recently, it was shown that LAG-3 is preferentially expressed on CD56^dim^ NK cells after IFN-α treatment and acts specifically as a negative regulator of cytokine production without compromising cytotoxic function [93]. Another candidate, TNF receptor family member, OX40 is expressed on leukemic blasts in a substantial percentage of patients with AML and promotes activation and proliferation of T cells. OX40 ligand-induced signaling via OX40 promotes NK cell activation, cytokine production and cytotoxicity, and disruption of this interaction impairs NK cell reactivity against primary AML cells [94]. Thus, targeting this pathway may be beneficial for NK cell function against AML. Finally, the IL-1R8, a negative regulator of Toll-like and IL-R1 family signaling, is highly expressed on NK cells, and upregulated during NK cell maturation [95]. IL-1R8 inhibits NK cell activation and NK-mediated control of tumor growth and metastasis, highlighting its role as a checkpoint target in NK cell tumor immunity [95]. Therapies targeting these checkpoint molecules may improve NK cell function against cancer, particularly if used in conjunction with existing gold-standard treatments.

Additionally, combination therapies are currently needed to produce the greatest therapeutic benefit, as well as overcome resistance to checkpoint inhibition. In this regard, a recent study used combinations of DNA methyltransferase inhibitors with anti-TIGIT or anti-killer cell lectin-like receptor subfamily G member 1 antibodies and were able to reduce the formation of the tumor-exposed NK cell phenotype, an NK cell phenotype that is unable to promote tumor killing and is responsible for promoting cancer metastasis [96]. This may be useful in an adjuvant setting to prevent metastatic recurrence. Thus, the use of epigenetic modifiers in conjunction with checkpoint inhibitors are emerging as new therapeutic choices for cancer [97]. Further, BiKE and TriKE platforms are versatile and amenable to incorporation of unique combinations, allowing for robust NK cell expansion, NK cell-mediated cytotoxicity and NK cell survival in vitro. Presently designed BiKEs or TriKEs concurrently engage with CD16 and tumor antigens to induce NK-mediated ADCC. For example, CD16/CD19 BiKEs and CD16/CD19/CD22 TriKEs have effectively mediated NK cytotoxicity on a range of specific antigen-expressing targets and maintained functional stability after 24 and 48 h of in vitro culture [98]. It is possible to include checkpoint receptor-blocking single-chain variable fragments (scFvs) and various cytokines in these constructs to enhance NK cell-mediated antitumor responses [70]. Additionally, BiKEs and TriKEs have the advantage of increased biodistribution compared to mAbs, due to their significantly smaller size (50–75 kDA compared to 300–450 kDa of full-length antibodies), they are non-immunogenic and can be quickly engineered [70]. Effective design of these engagers could therefore enhance tumor recognition, reduce tumor escape and promote effector potency and specificity, which can improve long-term success in the management of aggressive and refractory malignancies [99].

### 3.3. Synthetic Compounds and Recombinant Proteins

Synthetic compounds such as nicotinamide (NAM), glycogen synthase kinase (GSK)-3 inhibitors and recombinant heat shock proteins have also been used to enhance the proliferation and/or killing capacity of NK cells (Figure 3).

#### 3.3.1. NAM

NAM is a form of vitamin B3 and a potent inhibitor of the NAD-related signaling pathways that control redox-sensitive enzymes, mitochondrial functions, cell metabolism and the production of energy and cell motility [100]. In vitro experiments have demonstrated that NAM supplementation promotes NK cell function, antibody-mediated killing, in vivo persistence and tissue trafficking in preclinical models and tolerance to oxidative stress. NAM (2.5–5 mM) enhances expansion (60–80-fold) of functional NK cells in feeder-free cultures (PB, cord blood or CD56 enriched cultures) and substantially upregulates the homing receptor CD62L in ex vivo-expanded human NK cells (Figure 3) [100,101]. These studies established that NAM-expanded NK cells enhance cytokine secretion against tumors, improve in vivo proliferation and homing to multiple organs including the bone marrow. Bachanova and colleagues reported the first-in-human phase I study of NAM-expanded donor NK cells for the treatment of relapsed/refractory NHL and MM [102]. In vitro culture of CD3-depleted mononuclear cells, post donor apheresis, with NAM and IL-15 resulted in a 40-fold increase in NK cells after 14–16 days of culture; with the expression of homing receptor CD62L increasing from 2.9% in apheresis to 21% in final product. NAM-NK was infused into 3 patients without any dose limiting toxicity. Response assessment at 2 months showed that 3 patients achieved complete metabolic remission. Flow cytometry analysis of PB showed proliferation of NAM-NK in blood between days 2 and 7 in all tested patients. Compared to host NK cells, donor NAM-NK cells demonstrated higher CD16 expression and enhanced proliferation [102]. As such, NAM could be a desirable addition in ex vivo NK cell expansion.

#### 3.3.2. Thalidomide and Thalidomide Derivatives

Immunomodulatory imide drugs (IMiDs) such as thalidomide and thalidomide derivatives, have the ability to indirectly augment NK cell cytotoxicity and proliferation via induction of IL-2 transcription and secretion by T cells and dendritic cells (Figure 4) [103,104]. Furthermore, during in vitro culture of NK cells, the percentage of CD56^+^ cells is higher in the presence of IMiDs relative to DMSO control cultures showing a role for these drugs in the proliferation of NK cells [103] (Figure 4). IMiDs trigger the activation of phosphoinositide 3-kinase, and subsequently the nuclear translocation of the nuclear factor of activated T cells 2 and activator protein 1. However, they do not activate extracellular signal-regulated kinase (ERK) or p38 mitogen-activated protein kinase, which mediate NK cell cytotoxicity and ADCC, thereby lending strength to the argument that IMiD-mediated NK cell activation is an indirect chain of events [103].

Cold-target inhibition assays, which show the susceptibility of cancer cells to effector-induced lysis, have demonstrated that iMiDs are able to induce NK cell-mediated lysis of MM cell lines and patient cancer cells. Moreover, in vitro depletion of CD56^+^ cells, but not of CD4^+^ or CD8^+^ cells, blocked drug-induced MM cell lysis, confirming that NK cells are the effectors mediating this response. Thalidomide itself enhances the proportion and cytotoxicity of NK cells in responding patients with MM [103]. Measurement of cytokines in the plasma from patients also showed increases in IL-2 and IFN-γ after thalidomide treatment. This was attributed to a direct effect on T cells, resulting in an increase in cytokine secretion, which subsequently augments NK cell number and function, and ultimately MM cell lysis [105]. Emerging data also suggest that combining IMiDs with tumor-specific mAbs may augment NK cell-mediated antitumor responses. Although IMiDs have shown remarkable efficacy in patients with cancer, the contribution of IMiD-mediated augmentation of NK cell function to these antitumor effects has yet to be established [104].

#### 3.3.3. GSK Inhibitors

GSK-3α/β has a multifaceted role in cancer, having been involved in either inducing or inhibiting tumor proliferation and progression, suggesting it has multiple targets in different conditions [106]. Recently in T cells, small-molecule inhibition of GSK-3 was shown to downregulate, both immune checkpoints PD-1 and LAG-3, through the upregulation of the transcription factor, T-bet. Further to this, increased antitumor immunity to B6 melanoma cells was observed using a combination of both GSK-3 inhibition and antibody blockade of LAG-3. GSK inhibition also reduces LAG-3 expression on NK cells, suggesting a similar mechanistic role for GSK-3 in both T cells and NK cells [107]. Targeting GSK-3 in combination with LAG-3 may therefore prove useful in cancer exhausted immune cells.

GSK-3 also negatively regulates multiple NK cell activating signals through immunoreceptor tyrosine-based activation motif (ITAM)-dependent and -independent pathways. Individual or combined engagement of the ITAM-independent receptors NKG2D and 2B4 induces inhibitory phosphorylation of GSK-3 at regulatory serine residues, while activation of the ITAM-dependent receptors CD16 and NKp30 inhibited the GSK-3β isoform specifically, potentiating the NK cell response. The extent of phosphorylation was closely correlated with the degree of NK cell activation [108]. NK cell cytokine production and cytotoxicity were consistently enhanced by the knockdown of GSK-3β or its inhibition with different pharmacological inhibitors, whereas inhibition of the GSK-3α isoform had no effect. In addition, NK cell function was augmented by the overexpression of a catalytically inactive form of GSK-3β [108].

In vitro studies have established that pharmacologic inhibition of GSK-3 in IL-15 ex vivo-expanded PB NK cells significantly upregulates CD57, a marker for late-stage NK differentiation and maturation. When these NK cells were in the presence of GSK-3 inhibitors and exposed to human cancer cells, this resulted in significantly higher production of TNF and IFN-γ, which translated into increased NK cell cytotoxicity and ADCC. There was also increased expansion and proliferation of these effector NK cells. Similar results were observed in vivo; in a xenogeneic model where luciferase-expressing SKOV-3 tumors were established in NSG mice. Adoptive transfer of GSK-3 inhibition conditioned NK cells displayed more robust and durable tumor control [109]. These studies indicate that the inhibition of GSK-3 can be used to increase the potency of NK cell function in vitro and in vivo, but given the many roles of GSK-3, its mechanism of action in different cancer types must be first elucidated.

#### 3.3.4. Hsp70

The expression of the stress-inducible heat shock protein 70 (Hsp70) on the cell surface of tumor cells acts as a recognition structure for NK cells [110]. Recombinant Hsp70 protein and synthetic Hsp70 peptide TKD (TKDNNLLGRFELSG; aa 450–463) are able to mimic surface-bound Hsp70 by increasing the cytolytic activity of NK cells [111] and initiate the migration of NK cells in a concentration-dependent, highly selective and chemokine-independent manner [112]. Incubation of PB lymphocyte cells with TKD and low-dose IL-2, enhances the cytolytic activity of NK cells against Hsp70 membrane-positive tumors, in vitro and in vivo [113]. The tolerability, feasibility, and safety of TKD-activated NK cells was assessed in a phase I clinical trial. In vitro analysis of patient peripheral blood lymphocytes revealed that after contact with TKD and low-dose IL-2, NK cells secreted high amounts of IFN-γ and marginal secretion of TNF-α. The apoptosis-inducing granzyme B, a surrogate for Hsp70 reactivity in NK cells, was detectable in the serum of all patients 1 day after reinfusion of TKD-activated cells. CD94, also known as the NK-activating receptor NKG2C, was also upregulated on NK cells of tumor patients after incubation with TKD. However, the expression of NCRs, NKp30 and NKp46 remained unaffected [113]. Addition of synthetic Hsp70 can therefore increase the cytotoxicity and manipulate chemotaxis of NK cells.

### 3.4. CARs

Immune cells can be equipped with CARs specific for tumor antigens. These constructs specifically target cells expressing the antigen of choice via a scFv and activate cells via an intracellular signaling domain. CAR-T cells have shown considerable success against liquid cancers, which is exemplified by the FDA approval of the CD19-specific autologous CAR products Tisagenlecleucel and Axicabtagene ciloleucel [114]. Despite this success, generating CARs for NK cells possesses challenges that may be addressed through the exploitation of NK cell function in specific tumor environments to generate more effective CAR constructs. CAR constructs are made up of the following main components: the ectodomain (comprised of a scFv and hinge), the transmembrane domain (TM) and the intracellular signaling endodomain (Figure 5).

#### 3.4.1. Ectodomains

The ectodomain portion dictates the antigen specificity of the CAR. CAR ectodomains targeting CD19 and CD20 have been used to treat B cell leukemia and lymphoma, CD138 for myeloma, CD33 and CD7 for AML, members of the human EGFR family for breast cancer and the disialoganglioside GD2 for neuroblastoma [115,116]. With advancing technology, dual CARs that have ectodomains targeting two different moieties or different antigens on the same target cells are fast becoming mainstream. One approach generating interest is multiple-antigen targeting, which aims to increase specificity, capture a variety of tumor clones and reduce antigen-negative relapse. An example of this approach is the targeting of EGFR and its mutant form EGFRvIII, which are overexpressed in a large proportion of glioblastomas (GBMs). Genßler et al. generated variants of NK-92 cells that express CARs carrying a composite CD28-CD3ζ domain for signaling and scFv antibody fragments recognizing either EGFR, EGFRvIII, or an epitope common to both antigens [117]. In vitro analysis revealed high and specific cytotoxicity of EGFR-targeted NK-92 cells against established and primary human GBM cells, which was dependent on EGFR expression and CAR signaling. EGFRvIII-targeted NK-92 only lysed EGFRvIII-positive GBM cells, while dual-specific NK cells expressing a cetuximab-based CAR were active against both types of tumor cells. In immunodeficient mice carrying intracranial GBM xenografts expressing either EGFR, EGFRvIII or both receptors. Local treatment with dual-specific NK cells was superior to treatment with the corresponding monospecific CAR-NK cells. This resulted in a marked extension of survival without inducing rapid immune escape, as observed upon therapy with monospecific effectors. Their results demonstrated that dual targeting of CAR-NK cells reduces the risk of immune escape [117]. In fact, some clinical trials are ongoing to assess bi-specific CAR-T cell therapy efficacy of anti-CD19/CD22 (NCT03448393) and anti-CD19/CD20 (NCT03271515) to prevent recurrence caused by antigen loss [116], which could be translated into the NK platform too.

#### 3.4.2. Endodomains

CARs for CAR-NK cell therapy must be specifically designed with the unique characteristics of NK cells in mind. CAR-induced cytotoxicity and cytokine production is mediated through adapter protein signaling. In general, first-generation CAR-Ts typically have CD3ζ as their signaling domain, and second- and third-generation CARs sequentially contain other co-stimulatory molecules such as CD28, 4-1BB and CD314. NK-specific CAR constructs have largely focused on incorporating either DNAX-activation protein 10 or 12 (DAP10 or DAP12) as the activating domain or as a costimulatory domain alongside CD3ζ [118]. A CD19-targeting first-generation CAR, CD19-CD3ζ, was shown to outperform CD19-DAP10 CARs, and its activity was further improved by the addition of a 4-1BB co-signaling domain [119]. The inclusion of both DAP10 and CD3ζ signaling, together with the activating molecule NKG2D arms NK cells with better killing power for targeting many cancer types.

NKG2D-DAP10-CD3ζ CAR transduction of NK cells increases NKG2D surface expression, infers more cytotoxicity against leukemia and solid tumor cell lines compared to mock-transduced cells, triggers secretion of IFN-γ, GM-CSF, IL-13, MIP-1α, MIP-1β, CCL5 and TNF-α, and induces a massive release of cytotoxic granules, which persist after 48 hours of continuous stimulation [120]. These cells also had considerable in vivo antitumor activity in a mouse model of osteosarcoma compared to control NK cells [120]. However, in the TME suppressive molecules such as TGF-β [121], may downregulate DAP10 subsequently limiting NK cell function. To overcome this suppression, studies have modified NK cells to contain a chimeric NKG2D receptor with the native molecule fused to CD3ζ [122,123]. More recently, this has shown that unlike endogenous NKG2D, the chimeric version is not susceptible to down regulation via the TME. When these cells were generated from patients with neuroblastoma they were able to kill autologous myeloid-derived suppressor cells capable of suppressing CAR-T function [123]. Thus, a combination therapy inclusive of these cells may be of benefit in tackling solid tumors.

Unlike DAP10 in first-generation CAR-NK cells, DAP12 outperformed CD3ζ in first-generation prostate stem cell antigen (PSCA) targeting CARs, both in vitro and in vivo [124]. An anti-PSCA-DAP12 CAR improved cytotoxicity of the NK cell line YTS against PSCA-positive tumor cells when compared with a PSCA CAR containing the CD3ζ domain. Notably, infused YTS NK cells armed with PSCA-DAP12 CARs resulted in complete tumor eradication in vivo.

The selection and placement of costimulatory domains can have different impacts on the cytotoxic function and ultimately persistence and safety. Thus, it is important to establish the function of different ectodomain and endodomain combinations. Li and colleagues performed the largest comparison of NK-specific CAR constructs in NK-92 cells, which evaluated four different transmembrane domains (CD16, NKp44, NKp46, and NKG2D) and four different costimulatory domains (2B4, DAP10, DAP12, and CD137) in various combinations with CD3ζ [125]. The most effective construct contained the NKG2D transmembrane domain and 2B4 co-stimulatory domain which was attributed to their leading effector cell proliferation and function.

2B4 also has strong co-stimulatory effects in CARs targeting CD19 or GD2 [114,125]. In vitro stimulated NK cells from healthy donors and leukemia patients were gene modified to contain CD19 or GD2-specific CARs with either CD3ζ or 2B4 domains, alone or combined. Chimeric 2B4 signaling alone failed to induce IL-2 receptor upregulation and cytokine secretion but triggered a specific degranulation response. However, the co-integration of 2B4 and CD3ζ significantly enhanced all aspects of the NK cell activation response to antigen-expressing leukemia or neuroblastoma cells, including IFN-γ and TNF-α secretion and release of cytolytic granules [126]. They also overcame NK cell resistance of autologous leukemia cells while maintaining antigen specificity. These data indicate that 2B4 signaling has a potent costimulatory effect in NK cells when used in CAR domains.

A major issue of infused NK cells is their relatively short in vivo persistence. As mentioned previously, it is now well established that while IL-2 and IL-15 can prolong NK cell survival, systemic administration of IL-2 and IL-15 has been associated with severe side effects in clinical trials [127]. Others are trying to tackle this problem by engineering CAR-NK cells themselves to express cytokines (See Section 3.1). For example, cord blood-derived CD19 CAR-NK cells were engineered to express IL-15 [48]. Along with efficient killing, these cells also produced their own soluble IL-15, which improved their function and allowed growth and survival over a period of 42 days. However, the infusion of high doses of these cells into Raji xenograft mouse models induced CRS and fatality in 4 mice [48]. Clearly, engineering cells to secrete cytokines may cause problems and does not appear to be the only solution required to improve NK cell persistence. A way of addressing this and other potential safety concerns may be the incorporation of a suicide gene within the NK cells [127]. Takeda have already incorporated the inducible caspase 9 (iCasp9) suicide gene in TAK-007 (Table 1) and MD Anderson Cancer Center have placed their cord blood-derived CD19 CAR-NK cells containing iCasp9 in clinical trials (NCT03056339, see Section 3.1).

#### 3.4.3. Safety Features

NK-based therapy is generally considered a safer alternative to CAR-T therapy because of less occurrence and severity of side effects such as CRS. The cytokine production of NK cells is considered to be safer because the cytokines released are mainly IFN-γ and GM-CSF and once infused into patients CAR-NK cells are short lived, thereby reducing chances of other side effects on exposure. Several clinical studies have shown that NK cells do not cause GvHD [116] and safety and efficacy of NK cell therapies for hematological tumors such as AML are well established [128]. New clinical trials targeting hematological malignances are attempting to improve treatment response and NK cell number and dose. For example, a clinical trial showed that CAR NK-92 cells can be infused at doses up to 5 billion cells per patient without causing significant adverse effects in patients with relapsed or refractory AML [129]. There are many ongoing clinical trials for both hematological and solid tumors that have not yet reported safety outcomes, but are elegantly reviewed in Liu et al. [128]. Regardless, the potency of CAR-NK cells and also the method of generation and expansion of these edited cells warrant safety features as discussed below.

#### 3.4.4. iCasp9 and Other Inducible Safety Features

An ideal suicide gene would guarantee the safety of cellular therapies by permanently removing the cells responsible for any toxicities. To increase safety of CAR-T cell therapies, vectors designed to deliver the CAR can be engineered to co-express suicide genes to allow removal of CAR-T cells in case of uncontrollable severe adverse events. Examples of clinically available suicide gene strategies include the herpes simplex virus thymidine kinase (HSV-TK) suicide gene, which makes cells sensitive to ganciclovir-induced cytotoxicity or the iCasp9 gene cassette, leading to rapid caspase-mediated apoptosis of expressing cells upon application of a synthetic inducer of dimerization, such as AP1903 or AP20187 [130]. Currently, phase I clinical trials for the treatment of sarcoma and neuroblastoma utilizing GD2 CAR-T cells (NCT01822652) containing suicide genes, are ongoing [131]. The effectiveness of four suicide gene strategies have been compared in vitro using Epstein Barr virus cytotoxic T cells genetically modified to express HSV-TK, iCasp9, mutant human thymidylate kinase (mTMPK), or human CD20 codon optimized suicide gene. In this study, activation of HSV-TK, iCasp9, and CD20 ultimately resulted in equally effective destruction of transduced T cells. However, while iCasp9 and CD20 effected immediate cell-death induction, HSV-TK-expressing T cells required 3 days of exposure to ganciclovir to reach full effect, and mTMPK-transduced cells showed lower T cell killing at all time points [132].

The source of the suicide gene is also an important component of the strategy design. For example, in contrast to the iCasp9 suicide gene, which is almost completely human derived, viral-derived systems, such as HSV-TK, proved immunogenic in immune-competent patients with limited persistence of HSV-TK cells [133]. Therefore, there is a preference for using a modified human iCasp9, fused to a human FK506 binding protein to allow conditional dimerization using a small-molecule pharmaceutical. A single 10 nM dose of synthetic dimerizer drug induced apoptosis in 99% of transduced cells selected for high transgene expression in vitro and in vivo [134]. This system consists of human gene products with low potential immunogenicity, the administration of dimerizer drug has no effects other than the selective elimination of transduced T cells and that iCasp9 maintains function in T cells overexpressing antiapoptotic molecules [134]. Although mostly implemented in CAR-T cells, such safety features should perhaps also be incorporated in CAR-NK cells.

#### 3.4.5. Masked CARs

NK cell safety may also be improved by using masked CARs, a technique currently being investigated in T cell safety. CAR masking involves the use of a peptide that blocks antigen-binding sites and a protease junction. When protease-cleaved junctions are detached from masking peptides in the TME, CAR-T cells are kept active in local tumor sites [135]. One example is an EGFR-specific masked CAR, which consists of a masking peptide that blocks the antigen-binding site and a protease-sensitive linker. Proteases commonly active in the TME can cleave the linker and disengage the masking peptide, thereby enabling CAR-T cells to recognize target antigens only at the tumor site. In vitro masked CARs show dramatically reduced antigen-binding and antigen-specific activation in the absence of proteases, but normal levels upon treatment with certain proteases, along with comparable antitumor efficacy to that of unmasked CARs in vivo [135]. The effectiveness of this approach clinically is obviously dependent on the level of the required proteases, which is likely to differ within and between the TME.

#### 3.4.6. Oxygen-Sensitive CARs

Another strategy to generate CAR-T cells that are effective at the tumor site (thereby potentially reducing off-target effects), is the use of oxygen-sensitive CARs. By combining an oxygen-sensitive subdomain of hypoxia-inducible factor 1 with a CAR structure, CAR-T cells can subsequently respond to a hypoxic environment, a hallmark of many solid tumors. Juillerat et al. used a double-input approach within a single CAR, by including both antigen and hypoxic recognition. Under hypoxic conditions, they observed increased CAR surface expression and improved cytolytic properties [136]. Unlike those focused on improving CAR-T cell cytotoxicity and proliferation through design of second- and third-generation CAR molecules, this approach showed that the TME can induce innately self-deciding CAR-T cells with the ability to inactivate in presence of hypoxia. However, additional studies are required to confirm the validity of this approach. This study was not only the first proof-of-concept of a stand-alone use of the engineered oxygen-sensitive CAR, but an amenable platform for multi-receptor Boolean logic gate application of “AND” and “NOT” gates to further enhance the control of engineered cells [136].

#### 3.4.7. Combinatorial Targeting

The Boolean logic gates of “AND”, “OR”, and “NOT” have been applied to strategies for developing multi-antigen-targeting CAR-T cells that avert tumor antigen escape and reduce on-target, off-tumor toxicities [137]. Combinatorial targeting can be achieved by splitting the activation and co-stimulation across two receptors (logic AND gate) or by combining activating and inhibiting CARs (logic NOT gate) based on the concepts of Boolean logic. On-target, off-tumor effects can be minimized by adding AND gated circuits requiring both antigens to be present for CAR activation, or NOT gated circuits which will activate in the presence of one antigen only if the other is not present. Targeting one antigen OR another can eradicate multiple clones and reduce antigen-negative relapse, and can be achieved by either infusing two separate populations of CAR-T cells, transducing two CARs into the same cell, or by the novel tandem CAR [138].

Morsut and colleagues, as well as Roybal and colleagues improved the AND gate combinatorial detection of multiple antigens by engineering Notch-based receptors that function independently from a CAR-activated T cell receptor (TCR) pathway [139,140]. The design of combinatorial antigen targeting is taking advantage of the synthetic Notch receptors (synNotch), a new class of modular receptors comprising an extracellular recognition domain; the transmembrane “core” domain; and the intracellular transcription domain that can be cleaved and released by a transcriptional activation domain translocating to the nucleus and regulating transcription upon ligand engagement. Furthermore, CAR-T cells have been designed to activate when SynNotch and CAR ligands are co-expressed on tumors, a strategy that spatially identifies and effectively controls CAR-T cell activation [139,140]. Using combinatorial or dual targeting by CARs can increase their safety profile. Through the incorporation of the synNotch platform, Roybal et al. constructed two combinatorial antigen recognition T cell circuits [CD19 synNotch/mesothelin CAR and green fluorescent protein synNotch/CD19 CAR] and demonstrated that these receptors could conditionally express CARs specific for a second antigen in the presence of the first antigen-specific for the synNotch receptor [140].

Another form of combinatorial targeting is the use of inhibitory CARs (iCARS). If dual antigens are simultaneously expressed on healthy cells rather than on tumor cells, the combination of inhibitory receptors specific for the antigen present on normal but not on tumor cells will protect the normal cells from a CAR-T cell-mediated attack because of negative signaling conferred by iCARs. Fedorov et al. pioneered an anti-prostate-specific membrane antigen iCAR, carrying intracellular tails of CTLA-4 or PD-1 and tested whether these receptors have the ability to block TCR- or CAR-driven T cell functionality in vitro and in vivo [141]. This study demonstrated that the iCAR can inhibit TCR or CAR responses in an antigen-restricted manner. T cell stimulation experiments showed that iCAR-mediated inhibition was reversible which ensured that most of the T cells’ previous engagement of iCAR can retain functionality, although some T cells could be anergized over time [142].

A potential disadvantage of combinatorial antigen-targeting strategies is the possibility of target antigen modulation leading to tumor escape, with increasing probability of escape with the requirement of additional antigen targeting. The decrease in the target antigen could potentially render the engineered CAR system either partially or completely unresponsive or non-protective. Additionally, the selection of multiple suitable targets might be complicated due to their different expression levels [136].

### 3.5. Genetic Ablation to Improve Function

Knockout (or knockdown) of regulators in the signaling pathway that restrict NK cytotoxicity and/or expansion can also aid in augmenting NK function.

#### 3.5.1. CIS

The suppressor of cytokine signaling (SOCS) proteins (CIS, SOCS1-7) are intracellular negative regulators that are often induced to limit cytokine signaling in a classic negative feedback loop [143]. The cytokine-inducible SH2-containing protein (CIS, encoded by CISH) is a negative regulator of IL-15 signaling and is increased in NK cells upon cellular activation and thus functions as a key suppressor of IL-15 signaling in NK cells [60]. IL-15 binding induces activation of the Janus kinase (JAK)1 and JAK3 tyrosine kinases, recruitment and activation of signal transducer and activator of transcription 5 (STAT5), and transcription of STAT5-target genes, which drive survival and proliferation of NK cells and encode cytotoxic agents such as granzymes [144]. Systemic delivery of IL-15 to patients has toxic side effects. Thus, new approaches that deliver IL-15 more precisely to the target cell or specifically render the target cell more responsive to IL-15 are appealing avenues for investigation (see Section 3.1).

CIS is an important intracellular checkpoint in NK cell-mediated tumor immunity. Deletion of CISH results in elevated IL-2Rβ expression and increased activity and expression of JAK1 and JAK3/STAT5 pathways, with a corresponding increase in NK cell survival, proliferation and cytotoxicity [145]. Loss of CIS also results in heightened and prolonged IL-15-driven JAK–STAT signaling in NK cells. CISH^−/−^ NK cells are hypersensitive to IL-15 and, as a result, CISH^−/−^ mice are resistant to experimental tumor metastasis. In this regard, intravenous administration of B16F10 melanoma cells to CISH^+/+^ mice resulted in extensive metastatic nodule formation in the lungs by 14 days. In contrast, B16F10 metastatic nodules were largely absent from CISH^−/−^ mice [145]. Similar differences in lung metastasis were observed when using the RM-1 prostate cancer cell line. Depletion of NK cells or neutralization of IFN-γ, but not depletion of CD8^+^ T cells, rendered CISH^−/−^ mice susceptible to B16F10 metastasis, suggesting that CIS is involved in the negative regulation of NK cell activity and IFN-γ in this particular model [145].

However, despite the notion that NK cells control many hematopoietic malignancies, there can be a variable degree of CIS-dependent protection depending on the tumor model. CISH deficiency did not seem to have a major impact on disease progression in mouse models injected with the RMA-S or Vk12653 MM cell lines [145]. Combining blockade of the inhibitory receptor CD96 or administration of low-dose IL-15 with CIS inhibition could further boost NK cell activity and enhance protection against tumor metastasis [144]. The impacts of CISH deletion appear to be restricted to a subset of those tumors where NK cells are critical. NK cells critically regulated by IL-12, IL-18, and type I IFN are less likely to be influenced by CISH deletion. This may be the case in the TME where NK cells may engage directly with tumor cells or in crosstalk with antigen-presenting cells, and IL-12, IL-18 and type I IFN are influential [145].

Nevertheless, deletion of CISH in NK cells derived from iPSCs may be useful as off-the-shelf therapies (see Section 4) for a variety of cancer indications. Zhu et al. have recently developed human CISH-knockout NK cells using an iPSC-NK cell platform, which demonstrated increased IL-15-mediated JAK–STAT signaling activity, improved expansion and increased cytotoxic activity against multiple tumor cell lines when maintained at low cytokine concentrations [146]. They also displayed significantly increased in vivo persistence and inhibition of tumor progression in a leukemia xenograft model and mechanistically improved metabolic fitness which contributed to enhanced NK cell function. CISH^−/−^ iPSC-NK cells were shown to be phenotypically mature and have normal proliferative capacity under highly stimulated culture conditions exhibiting significantly increased cytotoxic granule release (CD107a expression) and IFN-γ production when stimulated. Analysis of cell signaling pathways demonstrated markedly increased phosphorylation of IL-15-stimulated JAK1 tyrosine phosphorylation, STAT3, and STAT5 in CISH^−/−^ iPSC NK cells. Activated NK cells upregulate the rate of glucose-driven glycolysis and oxidative phosphorylation to provide energy to drive the pathways necessary for their effector functions. This suggests that the increased IFN-γ production and cytotoxic activity of CISH^−/−^ iPSC-NK cells were supported by increased glycolysis and oxidative phosphorylation [146].

Together, these studies demonstrate that CIS plays a key role in the regulation of human NK cell metabolic activity and thereby modulates antitumor activity.

#### 3.5.2. CDK8

Cyclin-dependent kinase 8 (CDK8) belongs to the family of transcription regulating CDKs. CDK8 regulates transcription downstream of cytokines utilizing the JAK–STAT pathway, thereby also interfering with and regulating immune responses [147]. CDK8 deletion in mice does not affect the frequency or total number of NK cells in the bone marrow and has no significant effect on the maturation of these cells. Therefore, this indicates that CDK8 is dispensable for the growth and development of NK cells. CDK8 deleted NK cells display markedly increased perforin expression and increased cytotoxicity in vitro [147]. This is because CDK8 mediates STAT1–S727 phosphorylation which inhibits NK cell cytotoxicity. The knockdown of CDK8 in primary NK cells reduces STAT1-S727 phosphorylation in mice [148].

#### 3.5.3. DGK

Diacylglycerol kinases (DGKs) are physiological regulators of T and NK cell development, differentiation and function. They negatively regulate diacylglycerol (DAG) signaling by phosphorylating DAG into phosphatidic acid [149]. DAG leads to downstream activation of ERK, NFκB, and AKT pathways mediated by protein kinase C and Ras activating protein [150]. Cytotoxicity and production of IFN-γ, which are controlled by the ERK pathway, are the most important effector activities required for tumor elimination. Thus, control of the ERK pathway ultimately determines T and NK cell antitumor activity.

There are 10 different isoforms of DGK comprising five different classes, each with specific structural motifs, tissue localization and functional roles. Of these variants, DGKα and DGKζ are dominantly expressed in T cells [151,152].

The inhibitory role of DGKs in CAR bearing T cells in the TME is exemplified when CAR-Ts are isolated from mesothelin-expressing flank tumors and evaluated for effector functions and status of inhibitory pathways. These CAR-T cells become rapidly hypofunctional with upregulation of intrinsic T cell inhibitory enzymes (DGKα, DGKζ, SHP-1) as well as expression of other surface co-inhibitory receptors [153]. Hypofunctional CD8^+^ tumor-infiltrating T cells from clear cell renal carcinoma also displayed upregulation of DGKα [154]. However, functionality was restored upon use of the commercial DGK inhibitor R-59022. Prinz et al. also rescued NK cell function in clear cell renal carcinoma by blocking DGK-mediated diminishing of ERK pathway using R-59022 [155]. Since R59022 is a relatively selective DGKα inhibitor, restored degranulation capacity of tumor-infiltrating lymphocytes (TILs, such as CD8^+^ T cells and NK cells) was attributed to DGKα. Of note, the level of degranulation of TILs in the presence of DGK inhibition was not higher than that observed with non-tumor-infiltrating lymphocytes, indicating that DGK inhibition can restore suppressed degranulation but does not augment degranulation beyond NK or T cell intrinsic response efficacy [149]. This finding helps alleviate concerns about potentially unleashing undesirable autoimmunity through DGK inhibition, which is an important issue when considering potential targeting of DGK in a clinical setting. Collectively, these results suggest that DGK inhibition may not only prevent development of unresponsiveness but may also be able to restore activity of suppressed immune cells.

Existing methods for DGK inhibition are challenging. For example, there are no specific DGKζ inhibitors available to date [156]. Commercial DGK inhibitors R59022 and R59949 are only relatively selective for DGKα and could therefore also be inhibiting a broad range of DGK isotypes [157]. Additionally, their chemical structure is analogous to ritanserin (a serotonin receptor antagonist) and have demonstrated higher binding affinity to serotonin receptors [158]. This challenges the hypothesized biochemical mechanism of improved TIL efficacy previously observed with these inhibitors. Better inhibitors are underway; such as a more specific inhibitor for DGKα [159] or a recently patented new class of promiscuous DGKα/DGKζ inhibitors [156], but DGK isoform-specific knockout models are of increasing interest in the field.

Both DGKα and DGKζ knockouts, either individually or simultaneously, have shown improvement in T cells. DGKζ^−/−^ T cells showed increased proliferation, enhanced IL-2 production and enhanced Ras-ERK activation upon TCR stimulation than wild-type (WT) T cells both ex vivo and in vivo [160,161,162]. In addition to this improved in vitro and ex vivo performance, DGKζ^−/−^ mice rejected various tumors more efficiently than WT mice. DGKα^−/−^ T cells also resulted in enhanced Ras-ERK activation and increased proliferation post TCR stimulation, but unlike DGKζ^−/−^ T cells they displayed normal phosphatidic acid production, which highlights the downstream molecular differences between the two isoforms [163,164]. DGKα^−/−^ T cells displayed a weaker increase in cytokine production and ERK phosphorylation compared to DGKζ^−/−^ T cells in vitro. However, adoptive transfer of either DGKα^−/−^ or DGKζ^−/−^ CAR-T cells yielded similar increases in efficacy compared to WT T cells in the treatment of murine mesothelioma [163,165]. Double-isoform knockout of DGKαζ using CRISPR/Cas9 has generated CAR-T cells that are less sensitive to hypofunction and have increased antitumor effects [166]. Taken together, these studies indicate that although the ideal target isoform is still ambiguous, DGK knockout reverses exhaustion and results in more functional T cells.

Genetic ablation of DGKζ leads to hyperresponsive NK cells in a cell intrinsic and developmentally independent manner. Interestingly, DGKζ^−/−^ but not DGKα^−/−^ NK cells are hyperresponsive to activating receptor stimulation [167]. Upon activation of three distinct activating receptor families (ITAM-dependent: NK1.1, Ly49D; costimulatory-like: NKG2D; SAP-dependent: 2B4), an increased fraction of DGKζ knockout NK cells degranulated and produced IFN-γ compared with WT NK cells. In contrast, there was no significant difference between DGKα^−/−^ mice and WT controls in terms of both degranulation and cytokine production, suggesting that DGKα does not contribute to limiting NK cell function. DGKζ^−/−^ NK cells exhibited increased pERK, but not pAKT or enhanced IkBa degradation, suggesting that increased pERK could be a major contributor to the enhanced NK cell function seen in DGKζ^−/−^ NK cells. Taken together, these data suggested that DGKζ deficiency enhances NK cell responsiveness to activating receptor stimuli in an NK cell-intrinsic and developmentally independent manner, by affecting ERK, AKT, and NF-κB pathways. Although the loss of DGKα does not seem to influence NK cell effector function and thus may not negatively regulate signaling downstream of NK cell activating receptors, a more in-depth study into its other roles could potentially demonstrate a role for DGKα in NK cell function [167]. Taken together, CRISPR/Cas9-induced knockout of DGK isoforms and other checkpoints may be another promising tool in battling tumors using NK cells.

## 4. Off-the-Shelf NK Cells for Cancer Immunotherapy

The next generation of NK cell products will ideally be available off the shelf. However, development of off-the-shelf NK cells would ideally require that they can be manufactured and modified (genetically and as described above) prior to need, cryopreserved and then infused on demand. Before this can be realized, there are several considerations that need addressing including the source of cells.

### Sources of NK Cells

Whilst it has been extensively demonstrated that NK cells are potent cytotoxic cells in vitro, in vivo and clinically, sourcing sufficient numbers of NK cells for clinically relevant doses remains challenging. NK cells for clinical use are primarily derived from PB. However, they are also available as immortalized cell lines such as NK-92 or can be differentiated from stem cell sources. Autologous NK cells derived from the patient are difficult to expand and often of poor quality following extensive lymphodepleting chemotherapy. Furthermore, these cells can display limited antitumor effects for a range of malignancies due to KIR-HLA matching-induced “self-tolerance” [168]. Therefore, ensuring a mismatch between donor NK KIR and recipient HLA is imperative for enhancing the resultant tumoricidal effect. Whilst anti-KIR antibodies may also block KIR-HLA matching and thereby increase NK function [56], allogeneic NK products offer an easier approach to facilitate HLA-KIR mismatch without having been previously exposed to chemotherapy.

The immortalized NK cell lines NK-92, KHYG-1, HANK-1, NKG, NK-YS, YTS, YT, NK3.3 and NKL have been used in preclinical studies. However, to date, only NK-92 cells have been approved for clinical use due to observed clinical benefits and minimal side effects [19,169]. Of note, from the 19 (as of March 2021) current clinical studies assessing CAR-NK cells, 5 are NK-92 based, which is indicative of successful preclinical investigations [170]. Even though NK-92 cells provide a flexible NK-based immunotherapy platform, they have certain shortcomings. Since NK cells have a short life span and have limited in vivo expansion, this ultimately may require repeated patient treatment to maintain or elicit the desired clinical effect. NK-92 cells originate from a non-Hodgkin lymphoma, therefore they must undergo γ-irradiation prior to infusion to minimize the risk of secondary lymphoma formation in patients [171]. This, however, may further limit their function and persistence in vivo. In line with their limited expansion kinetics and malignant origin, Tang et al. have recommended the use of CAR-NK-92 cells to treat slow progressing malignancies such as MM and inclusion of suicide genes as a safety measure in lieu of irradiation [129]. Nevertheless, the possible tumorigenicity of NK-92 cells remains a major concern.

A recent study by Oberoi et al. [42] showed that healthy donor PB can serve as a valuable starting material for creating hematopoietic stem and progenitor cells (HSCs), despite earlier studies indicating poor and donor-dependent differentiation [172,173,174,175,176]. This was due to a refined approach to differentiation and expansion protocols that utilized optimized cytokine cocktails and expansion enhancement with K562 feeder cells, co-expressing 4-1BB ligand and mbIL-15 and IL-21 [42] (see Section 3.1). As healthy PB is readily available and easier to access compared to umbilical cord blood (UCB) and human embryonic stem cells (hESCs), this protocol may serve as an economically viable option for NK cell generation.

An important and rapidly emerging source of NK cells are HSCs. Functional NK cells can be differentiated from HSCs derived from UCB, iPSC lines and hESCs [177,178,179]. Furthermore, NK cells derived from HSCs do not need irradiation prior to dosing. Mice treated with hESC-NK cells showed complete clearance of subcutaneous K562-induced tumors, with some animals remaining tumor free for 8 weeks after just one treatment injection. A significant reduction in metastases was also observed after treatment with the hESC-NK cells. In contrast, UCB-NK cells were shown to be less effective, with significantly more cancer growth [179]. Although further studies are required to confirm these comparative observations, ethical concerns related to the use of hESCs and the low initial number of progenitor cells in multipotent UCB are relevant limitations [179].

iPSCs, however, can potentially undergo unlimited expansion in vitro without losing pluripotency and therefore provide a potentially endless supply of high purity (>97%) NK cells [177]. Accordingly, it is thought iPSC-derived NK cells have the potential to be a potent off-the-shelf immunotherapy allowing for lower-dosage treatments and flexible administrations. Phenotypically, these cells have similar activating receptor profiles to PB NK cells, but negligible KIR expression [179,180,181,182]. The reduced inhibitory receptor expression may explain the observed higher cytotoxicity of iPSC-derived NK cells. It will be important to further elucidate how KIRs contribute and affect the function and specificity of UCB and iPSC NK cells. In this regard, Goldenson et al. recently showed NK cells expanded directly from UCB (UCB56) had higher KIR expression than NK cells generated from CD34^+^ cells derived from UCB (UCB34), with UCB56 NK cells exhibiting more cytotoxicity. They also found that iPSC lines derived from NK cell populations had variable KIR expression, with some lines generating NK cells with high levels of KIR, and some with low levels. However, no difference in cytotoxicity was observed between these iPSC lines. They investigated the importance of the aforementioned differences of KIR expression by measuring cytotoxicity against targets expressing or lacking KIR ligands using a cell line system that expressed HLAs. The presence of these ligands did not alter the difference in killing activity between UCB56 and UCB34 cells, or iPSCs suggesting the widely accepted HLA recognition role of KIRs is not occurring in their system, as HLA presence on the targets should have decreased killing capacity of high expressing KIR^+^ NK cells. The authors suggest that this may be attributed to feeder-based expansion which drives differentiation at the same time it attenuates inhibitory signaling, but other influences may also be possible [183].

CAR-engineered iPSC NK cells have successfully targeted human tumors in preclinical studies and currently there are a number of ongoing clinical trials for cancer immunotherapy using engineered iPSC NK cells [50,125,128]. Furthermore, differentiation strategies for iPSC to NK cells are adaptable as they can allow for focus on obtaining a certain phenotype and function for the resulting product. Thus, the high proliferation capacity and unlimited expansion potential, as well as the ease of gene editing and modification of iPSCs make these cells a candidate for developing libraries composed of off-the-shelf haplotype-specific cells for treating a range of diseases [128].

## 5. Conclusions 

NK cells, with their panel of multiple anticancer receptors and better safety profile, are emerging as crucial candidates for adoptive cellular immunotherapy for the treatment of cancer. However, significant roadblocks prevent their widespread clinical translation. As discussed above, NK cells comprise only 10% of the normal PB population, even less for patients undergoing chemotherapy which often causes lymphopenia, making NK cell extraction from patients very ineffective and invariably of compromised function. Another shortcoming is their reduced in vivo persistence [184].

Allogeneic NK cells, such as iPSC-derived NKs (iNKs) may provide a rational alternative. Given that iNK cells are not extracted from PB and therefore contain no contaminating T or B cells, there will be negligible risks of GvHD or a cytokine storm. Key questions still remain: could in vivo persistence be increased? Is it more desirable to increase iNK function to yield short-lived, potent killer cells, which also reduces the extent of side effects? How could genetic engineering, particularly CRISPR/Cas9 technology play a transformative role in generating these potent cells? If so, manipulation at what stage of the NK cell ontogeny would ensure successful and stable genome editing?

Engineered NK cells as immunotherapeutic agents are still in their infancy. There is an increasing body of evidence describing the success of CRISPR technology in NK cells [185]. Since mature NK cells have low persistence in vivo, the stable integration, expression and resultant effect of the edited gene has a very limited time to display, if at all. To allow for stable integration, stem cells could be edited and differentiated into functional NK cells. iNK functions could be modulated by manipulating their own natural cytotoxicity receptors, such as incorporating an antigen-specific scFv-based domain into them. In fact, observations that show NK-CARs function better when natural NK-signaling mechanisms are utilized [125] and that T cell receptor fusion constructs successfully employ the natural TCR signaling subunit of T cells [186] pave the way for novel receptor engineering of existing natural NK receptors. Perhaps this technique could allow for targeted and successful activity of adoptive immunotherapy into solid cancers. NK cells have already displayed in vitro and in vivo cytotoxicity in a number of ovarian cancer models such as SKOV-3 and MA-148 and could therefore be of clinical significance, particularly after appropriate editing to assist in trafficking and function [177,187].

## Figures and Tables

**Figure 1 cells-10-01058-f001:**
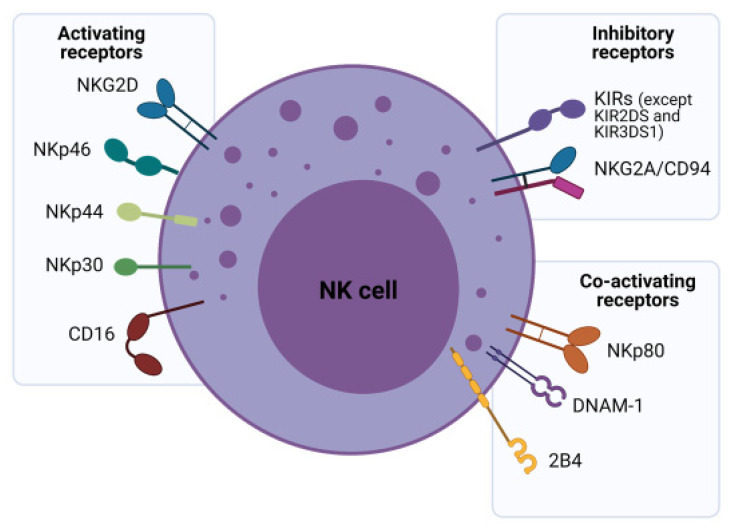
The major activating, co-activating and inhibitory receptors expressed on the surface of NK cells. Abbreviations: 2B4, also known as CD244; CD, cluster of differentiation; DNAM-1, DNAX accessory molecule-1; KIR, killer cell immunoglobulin-like receptors; NKG2, also known as CD159; NKp30, 44, 46, 80, natural cytotoxicity receptors.

**Figure 2 cells-10-01058-f002:**
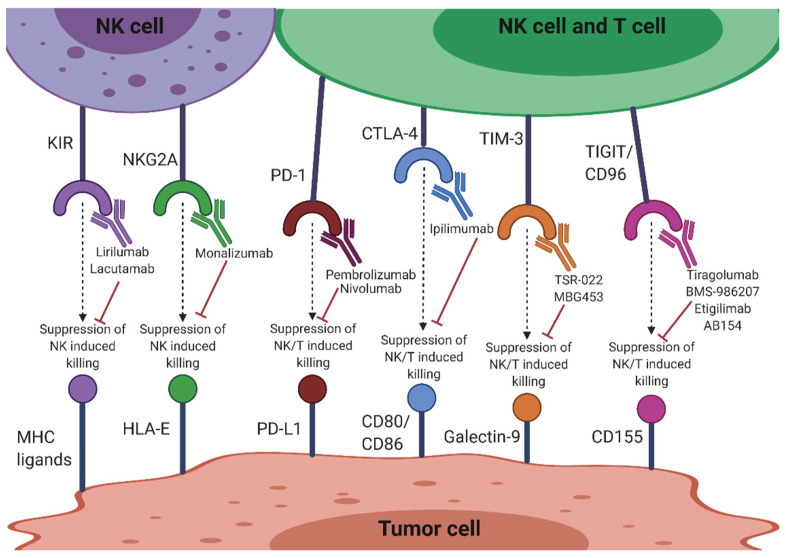
Checkpoint inhibitor targets in NK cells and T cells and their respective blocking antibodies under investigation for therapeutic efficacy and/or clinical use for cancer. Blocking antibodies act on their respective targets on NK cells and T cells to potentiate tumor cell killing. Abbreviations: CD, cluster of differentiation; CTLA-4, cytotoxic T-lymphocyte-associated protein 4; HLA-E, human leukocyte antigen-E; KIR, killer cell immunoglobulin-like receptors; MHC, major histocompatibility complex; NK, natural killer; NKG2A, CD94/NK group 2 member A; PD-1, programmed cell death protein 1; PD-L1, programmed death-ligand 1; TIGIT, T cell immunoreceptor with Ig and ITIM domain; TIM-3, T cell immunoglobulin and mucin domain-containing protein 3.

**Figure 3 cells-10-01058-f003:**
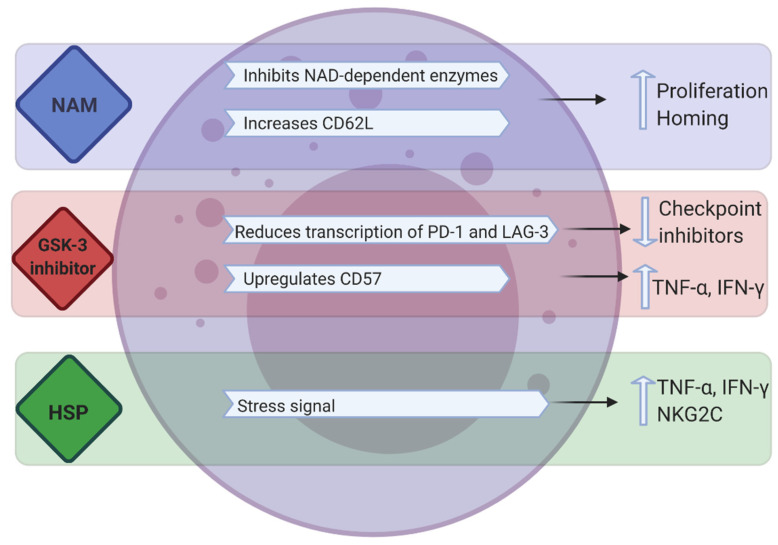
Effects of NAM, GSK-3 inhibition and HSPs on NK cell function. NAM induces the proliferation and homing of NK cells, while GSK-3 inhibition reduces checkpoint inhibitor transcription. HSPs are endogenous stress signals that induce an upregulation of NKG2C and both GSK-3 inhibition and HSPs induce a rise in TNF-α and IFN-γ levels. Abbreviations: CD, cluster of differentiation; HSP, heat shock protein; IFN-γ, interferon-γ; LAG-3, lymphocyte-activation gene 3; NAD, nicotinamide adenine dinucleotide; NAM, nicotinamide; PD-1, programmed cell death protein 1; TNF-α, tumor necrosis factor α.

**Figure 4 cells-10-01058-f004:**
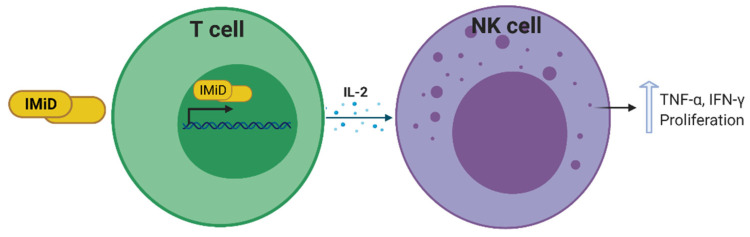
IMiDs mediate indirect augmentation of NK cell function. IMiDs act on IL-2 transcription in T cells to induce the production of TNF-α and IFN-γ and the proliferation of NK cells. Abbreviations: IFN-γ, interferon-γ; IMiD, immunomodulatory imide drugs; IL-2, interleukin-2; TNF-α, tumor necrosis factor-α.

**Figure 5 cells-10-01058-f005:**
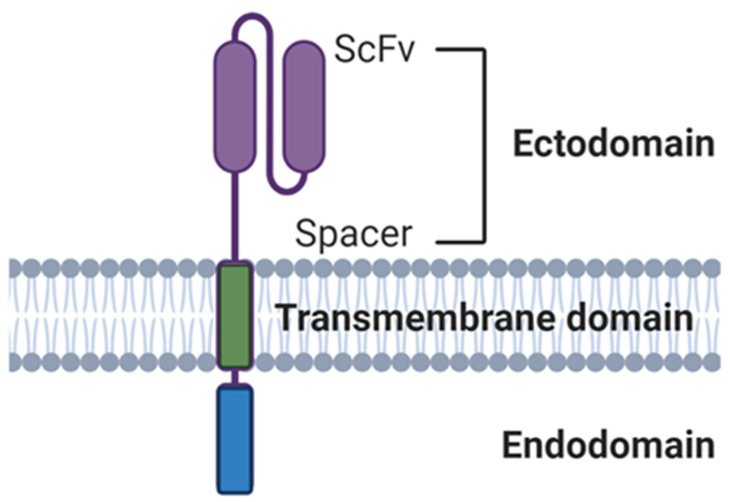
Components of a first-generation CAR. A CAR is made up of the ectodomain (composed of a scFv and hinge), the transmembrane domain and the intracellular signaling endodomain. Abbreviations: scFv; single-chain variable fragment.

**Table 2 cells-10-01058-t002:** mAbs targeting TIGIT for immunotherapeutic use [84].

Antibody	Company	Clinical Study Progress	Combination Therapy
Tiragolumab	Genentech/Roche	Preclinical studies	In combination with antibodies to PD-1
BMS-986207	Bristol-Myers Squibb	Phase I/II study NCT02913313	Monotherapy or in combination with nivolumab (anti-PD-1) in advanced solid tumors
Etigilimab	OncoMed Pharmaceuticals	Phase I clinical trial NCT031119428	Monotherapy or in combination with nivolumab (anti-PD-1) in advanced solid tumors
AB154	Arcus Biosciences	Phase I clinical trial NCT03628677	Monotherapy or combined with AB122 (anti-PD-1) in advanced solid tumors

Abbreviations: mAbs, monoclonal antibodies; PD-1, programmed cell death protein 1; TIGIT, T cell immunoreceptor with Ig and ITIM domain.

## Data Availability

Not applicable.

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
