# Peer review of "Enhancing a Natural Killer: Modification of NK Cells for Cancer Immunotherapy"

_cells, 2021, doi:10.3390/cells10051058_

Round 1
Reviewer 1 Report
The manuscript by Rasa Islam et al. is generally well-written. I am confident that it falls within the scope of Cells. However, there are a few critical things that need to be addressed.
First, for “3.2 Therapeutic approaches to enhancement of NK cell function”. It would be nice to discuss BiKE and TriKE in this section.
Second, it would be great to expand the safety features (3.4.3) section since more NK-related clinical results are coming. The side effect and safety features could be interesting for researchers.
Reviewer 2 Report
The manuscript of Rasa Islamand et al. is dedicated to a literature review on recent approaches to application of NK cells for anticancer immunotherapy. This field of investigation is categorically important because the search of novel effective anticancer therapy is still surely an important biomedical problem. The significance of this review is beyond doubt, since the presented summary of literature data and their analysis give an objective integral picture of existing ideas on the indicated topic and highlight possible ways of practical use of the accumulated knowledge. These include information concerning use of exogenous compounds to uphold NK killing capacity, targeting immune-function checkpoints, and development of NK CARs to provide cancer specificity. It is important to note that NK cell applications could allow targeted and successful adoptive immunotherapy for solid cancers.
In general the review is well presented; the data are of considerable novelty and interest.
Several minor suggestions might improve the overall quality of the manuscript.
- Page 3. “…transformed cells (abnormal cells such as tumors) express a variety of stress-related ligands such as Fas, death receptor 5, MHC I chain related proteins MICA/MICB and UL16 binding proteins, which are able to activate NK cells via their activating receptors [8,12].” It is good known that heat shock proteins (especially HSP70) are also stress-related molecules expressed on the surface on many types of tumor cells.
- Page 11. “Synthetic compounds such as nicotinamide (NAM), glycogen synthase kinase (GSK)-3 inhibitors and heat shock proteins have also …” Exogenous HSPs used “to enhance the proliferation and/or killing capacity of NK cells” should be indicated as the recombinant proteins rather than synthetic compounds. Additionally, HSPs are the real endogenous stress signal for the immune system rather than they “… mimic stress signals …” (Figure 3).
- Page 14. “Synthetic Hsp70 protein and Hsp70 peptide TKD …” should be corrected to “Recombinant Hsp70 protein and synthetic Hsp70 peptide TKD …”.
- Page 22. “4.1. Sources of NK cells”. This paragraph should include human NK cell clones developed by a number of laboratories as a possible source of NK cells for adoptive anticancer immunotherapy.
